# The Interaction of Cyclic Naphthalene Diimide with G-Quadruplex under Molecular Crowding Condition

**DOI:** 10.3390/molecules25030668

**Published:** 2020-02-04

**Authors:** Tingting Zou, Shinobu Sato, Rui Yasukawa, Ryusuke Takeuchi, Shunsuke Ozaki, Satoshi Fujii, Shigeori Takenaka

**Affiliations:** 1Department of Applied Chemistry, Kyushu Institute of Technology, Fukuoka 804-8550, Japan; zoutt@che.kyutech.ac.jp (T.Z.); shinobu@che.kyutech.ac.jp (S.S.); ryasukawa@takenaka.che.kyutech.ac.jp (R.Y.); rtakeuchi@takenaka.che.kyutech.ac.jp (R.T.); sozaki@takenaka.che.kyutech.ac.jp (S.O.); 2Research Center for Bio-Microsensing Technology, Kyushu Institute of Technology, Fukuoka 804-8550, Japan; 3Department of Bioscience and Bioinformatics, Kyushu Institute of Technology, Fukuoka 820-8502, Japan; s.fujii@bio.kyutech.ac.jp

**Keywords:** cyclic naphthalene diimide, G-quadruplex, dilute condition, molecular crowding condition

## Abstract

G-quadruplex specific targeting molecules, also termed as G4 ligands, are attracting increasing attention for their ability to recognize and stabilize G-quadruplex and high potentiality for biological regulation. However, G4 ligands recognizing G-quadruplex were generally investigated within a dilute condition, which might be interfered with under a cellular crowding environment. Here, we designed and synthesized several new cyclic naphthalene diimide (cNDI) derivatives, and investigated their interaction with G-quadruplex under molecular crowding condition (40% *v*/*v* polyethylene glycol (PEG)200) to mimic the cellular condition. The results indicated that, under molecular crowding conditions, cNDI derivatives were still able to recognize and stabilize G-quadruplex structures based on circular dichroism measurement. The binding affinities were slightly decreased but still comparatively high upon determination by isothermal titration calorimetry and UV-vis absorbance spectroscopy. More interestingly, cNDI derivatives were observed with preference to induce a telomere sequence to form a hybrid G-quadruplex under cation-deficient molecular crowding conditions.

## 1. Introduction

Over the past two decades, G-quadruplex has been attracting increasing attention for its important and mysterious roles in regulating the enzyme work and gene expression [1,2]. G-quadruplex, formed in the telomere region, was considered an obstruction to interfere with telomerase elongating telomere DNA, and therefore inhibits the activity of abnormal active telomerase in the cancer cell, further accelerating cancer cell death [3,4,5,6].

Chemical molecules, which can recognize and stabilize G-quadruplex structures that could be involved in G-quadruplex’s regulatory role in biology are considered. Their performance on stabilizing telomere G-quadruplex also gives them full potential to become a new type of anti-cancer drug [7,8]. Numerous G-quadruplex (G4) ligands have been reported [9], and some molecules, such as BRACO19 [10], pyridostatin [11], and naphthalene diimides [12], have been comprehensively adopted for biochemical and chemical biology studies. Our group has constructed cyclic naphthalene diimide (cNDI) and cyclic ferrocenylnaphthalene diimide derivatives as the cyclic shape could interfere with their interaction with dsDNA, thereby enhancing their selectivity for G-quadruplex [13,14].

However, the real cellular milieu is highly crowded with a diverse range of molecules, and all macromolecules in the physiological fluid media occupy 10%–40% of the total fluid volume. Such an excluded volume might be a major factor in driving the formation and even the transition of G-quadruplex, and the decreased water activity may also impair the binding of some molecules for G-quadruplex [15]. In order to obtain more objective investigations about the G4 ligands binding to G-quadruplex, neutral polymers such as polyethylene glycol (PEG) are commonly adopted to mimic the cellular crowding environment, since PEGs do not interact with G-quadruplex (therefore, they do not affect the overall thermodynamics of the system), and the volume occupied by PEGs grows large as the amount increases [16].

However, introducing PEGs into the evaluation system also brings challenges to accurately determine the binding affinity. The increased viscosity also makes complete mixing of ligands with a G-quadruplex solution more difficult. Until now, only a few reports have been published investigating the binding performance under molecular crowding conditions [15,17,18]. In this study, we prepared three cyclic naphthalene diimide (cNDI) derivatives with different cyclic substituents and investigated their performance for recognizing and stabilizing G-quadruplex both in dilute and molecular crowding conditions. Isothermal Titration Calorimetry (ITC) technique was adopted for evaluating the binding affinity and thermodynamic characteristics, and to help elucidate the binding property under a different environment. Additionally, for an unbiased judgment, Scatchard plot analysis based on UV-Vis absorbance spectra was also conducted to further confirm the binding performance. Interestingly, in the crowding condition, the telomere sequence generally forms a parallel G-quadruplex with the assistance of K^+^, while cNDI derivatives could favor a hybrid telomere G-quadruplex formation within a K^+^-absent crowding condition.

## 2. Results

### 2.1. Synthesis of Cyclic Naphthalene Diimide Derivatives

Based on our previously reported cyclic naphthalene diimide with cyclohexane substituent **1** [13], we designed another two new cNDI derivatives, **2** with aromatic ring substituent, and **3** with a simple alkyl chain substituent. Chemical structures of these cNDIs are shown in Figure 1. **1** was considered to cause steric hindrance for NDI to intercalate into double stranded DNA, and thus favor G-quadruplex specific binding, **2** was expected with better G-quadruplex binding affinity for its aromatic group possibly interacting with DNA base thymidine, and **3**, the most simple cyclic substituent, was supposed to be a comparison. These cNDIs were prepared, purified by HPLC, and confirmed by MALDI-TOF-MS and NMR (Appendix A).

### 2.2. cNDI Derivatives Recognize and Stabilize G-Quadruplex under Molecular Crowding Condition

In this study, two DNA sequences, c*-myc* and Telomere G1, were adopted to investigate the interaction of cNDIs with a G-quadruplex structure. A total of 40% (*v*/*v*) PEG200 was adopted to mimic the cellular crowding environment, which performed a similar molecular crowding effect as the previously reported 40 wt% PEG200 [19]. The G-quadruplex formation was identified by circular dichroism (CD) measurement, with adding cNDI derivatives **1**–**3**, the intensity shift of cotton effect that derived from cNDIs stacking to G-quartets was considered (Figure 2, Appendix A). Telomere G1 formed a hybrid G-quadruplex structure in dilute condition, but preferred to form parallel G-quadruplex in molecular crowding condition [19]. The same phenomenon was identified here, and three cNDI derivatives all triggered CD cotton effect shift under both situations (Figure 2a,b). c-*myc* formed parallel G-quadruplex in both dilute and molecular crowding condition, and the CD signals at 264 nm decreased as cNDIs were gradually added to the solution of nucleotides. These identified that cNDIs could recognize G-quadruplex under a molecular crowding condition (Figure 2c,d).

Then, the stabilization effect of **1**–**3** for G-quadruplex was evaluated by CD based melting temperature measurement. Due to the limitations of the technique, the potassium ion (K^+^) concentration was optimized for each determination to turn the evaluation into an available range. For 1.5 µM telomere G-quadruplex, 100 mM K^+^ was adopted in a dilute condition, and 1 mM K^+^ was adopted in the molecular crowding condition. Under both conditions, **1**–**3** could obviously enhance the melting temperature of telomere G-quadruplex (Figure 2e,f), and **2** showed the strongest ability to stabilize telomere G-quadruplex. Melting temperature (*Tm*) of ***2****-telomere G1 complex* in a dilute condition was 79 °C (*telomere G1 only*: 65 °C; ***1****-telomere G1*: 73.5 °C; ***3****-telomere G1*: 70.5 °C), and in a molecular crowding condition 73.5 °C (*telomere G1 only*: 59 °C; ***1****-telomere G1*: 67 °C; ***3****-telomere G1*: 65.5 °C), following the order of **2** > **1** > **3**. For 1.5 µM c-*myc* G-quadruplex, 5 mM K^+^ was adopted in a dilute condition, further adding 3 eq of **1**–**3** could similarly enhance the melting temperature for more than 15 °C; in molecular crowding condition with 0.01 mM K^+^, although ΔTm values were relatively smaller than those under dilute condition, **1**–**3** could still stabilize the G-quadruplex structure (Figure 2g,h).

### 2.3. cNDI Derivatives Bind to G-quadruplex under Molecular Crowding Condition

ITC measurement was a common technique adopted to evaluate the binding affinity of chemicals for specific targets. Here, we compared the binding performance of cNDIs for telomere G1 and c-*myc* in dilute condition and molecular crowding condition. In a dilute condition, we obtained a clear fitting curve for each cNDI-G-quadruplex binding process, and the thermodynamic parameters are shown in Table 1. **1**, **2**, and **3** all showed high binding affinity to G-quadruplex with *K*_a_ than in the 10^6^ M^−1^ order, and *K*_a_ of **1** for c-*myc* was several folds stronger than **2** and **3**. Under the molecular crowding condition, during titrating cNDIs to DNA solution, the saturation rates were slightly slower, indicating a decreasing binding ability (Appendix A). However, compared to previous reports, the decrease of cNDIs’ binding affinity for G-quadruplex were generally less than three-fold, and *K*_a_ could still keep around 10^6^ M^−1^ order (Table 1).

Adding PEG200 to the ITC system may also cause problems for accurate measurement. We further performed a Scatchard plot analysis based on UV-vis absorbance measurement (Appendix A). As results revealed in Figure 3, consistent binding affinities were acquired from both techniques, except for telomere G1 with **1** in the dilute condition that showed some variation. Furthermore, although **3** showed various affinities for telomere G1 and c-*myc* under a dilute condition, in a molecular crowding condition, similar *K*_a_ for both telomere G1 and c-*myc* were obtained. This might be explained by telomere G1 driven to form a parallel stranded G-quadruplex in the molecular crowding condition, which possibly diminishes the structural-derived binding difference.

### 2.4. cNDI Derivatives Induced the Formation of Telomere Hybrid G-quadruplex under Cation-Deficient Molecular Crowding Condition

Potassium ion (K^+^) was commonly adopted for assisting telomere G1 to form a hybrid G-quadruplex in vitro dilute condition, while under molecular crowding condition, a parallel stranded telomere G-quadruplex structure was more favored. Herewith, with the absence of K^+^, CD spectra of 1.5 µM un-annealed telomere G1 and c-*myc* were measured by adding **1, 2**, or **3** from 0–15 µM within molecular crowding condition. For c-*myc*, the molecular crowding condition could drive the formation of parallel G-quadruplex. Further adding cNDIs stabilized the structure and induced CD signals at 264 nm shifting (Appendix A). While for telomere G1, CD spectra representing the mixture of parallel and hybrid G-quadruplex were observed [20]. With the increasing amount of cNDIs, the cotton effect around 290 nm increased gradually. Especially by adding **2** with more than five equivalents to G-quadruplex, the transferred CD spectra became more hybrid-like compared to the CD spectra obtained in a dilute condition with 100 mM K^+^ (Figure 4b,d). The melting temperatures of the final induced cNDIs (10 eq)-telomere G1 complex with the absence of K^+^ were also determined (Figure 4e), and a significantly enhanced stability was observed. ***2****-telomere G1 complex* also displayed the highest stability. Based on this, without cation ion, cNDIs were supposed that could induce telomere to form a hybrid G-quadruplex even under a molecular crowding condition.

## 3. Discussion

Cyclic NDI was previously reported as a new strategy to enhance the selectivity of NDI for G-quadruplex rather than dsDNA. Here, with **1** and other new cNDI derivatives **2** and **3**, we investigated the influence of molecular crowding condition on cNDIs towards recognizing and binding to G-quadruplex in detail. All cNDIs could stabilize both parallel and hybrid G-quadruplex structures, while **2** with the aromatic ring as cyclic substituent showed a slightly stronger ability to increase the melting temperature of telomere G-quadruplex in both a dilute condition and a molecular crowding condition. 

However, we did not observe obvious differences among these three chemicals toward binding G-quadruplex, and the binding affinity was also not significantly interfered with under a molecular crowding condition. Possible reasons could be addressed: 1) cNDIs bound to G-quadruplex through π–π stacking, compared to cationic ligands binding to G-quadruplex, π–π stacking interaction require fewer numbers of water molecule, and therefore, suffered less influence from the decreased water activity caused by molecular crowding condition [15]; 2) similar to previous reports [21,22], based on the thermodynamic parameters obtained from ITC measurement, Δ*H* values under molecular crowding condition were significantly negatively larger, which revealed that the binding was enthalpically driven, suggesting that the complex was stabilized by the formation of favorable interactions between cNDIs and G-quadruplex. However, the negative entropic contribution found for the binding suggested that the resulting complex was less flexible than the free quadruplex. After coordinating the enthalpy and entropy effect, the Gibbs free energy was not significantly changed, and hence the cNDIs retained a comparatively high binding affinity to G-quadruplex under a molecular crowding condition.

Regarding the ideal crowding agent to mimic the cellular condition, the most important criterion is that the agent should not interact with the system under testing, and thence highly concentrated cell extracts might not be recommended because of the high probability of specific interactions, hydrolyase activity, and the presence of denatured proteins. Crowding effects can also be caused by concentrations of hemoglobin or serum albumin, but they prevent the use of optical assay methods due to the color problem [23]. Synthetic crowding agents, including Ficolls, dextrans, and PEG, are widely used, but they may cause precipitation and preferentially interact with proteins or nucleic acids. The large PEG generates an area inaccessible to other molecules and increases the solution viscosity, while small cosolute molecules such as ethylene glycol and PEG200 do not act as obstacles, but effectively change the solution property [24]. In this study, to elucidate the influence of PEG200, cation-ion, and the annealing process to DNA structure, CD spectra of telomere G1 under different conditions were tested and are shown in Appendix A. Under K^+^-deficient molecular crowding condition, without annealing, telomere G1 was induced to form a mixture of a parallel and hybrid G-quadruplex, while with an annealing process, dramatically no detectable CD spectra could be obtained, and one reasonable consideration might be that PEG induced telomere G1 aggregation or some unknown structure formation during annealing. Previously, the Sugimoto group also mentioned that DNA homopolymer poly(dA) exhibited no signal with 20 wt% PEG8000, which might occur due to the DNA precipitation by the high molecular weight of PEG during the annealing process [25].

Characterizing from CD spectra, under stabilizing cation-deficient condition, TBA G-quadruplex was reported to form and considerably stabilize by 40 wt% PEG [26]. In this study, with 40% *v*/*v* PEG, the telomere sequence could form a mixture of hybrid and parallel G-quadruplex. While increasing the concentration of cNDIs, the CD signals around 290 nm were gradually enhanced, although the initial ratio of parallel and hybrid G-quadruplex showed some variations during each determination. By adding cNDIs, the generally similar tendency of CD spectra transformation could be finally obtained (Appendix A). Especially **2** showed the strongest performance on favoring hybrid G-quadruplex transformation, and ***2**-telomere G-quadruplex complex* exhibited higher stability than the other two in all condition tests here, indicating **2** might possess a preference for telomere G-quadruplex.

Although parallel-stranded G-quadruplex conformation may be the more favored structure in a biologically relevant environment [19], such a crowding condition induced telomere parallel G-quadruplex structure is not as stable as a c-*myc* (in molecular crowding condition, Tm of telomere with 1 mM K^+^ at 59 °C, Tm of *c*-*myc* with 0.01 mM K^+^ at 67 °C). Furthermore, sequences containing four tracts of three guanines with loops of variable length, various cation type, and concentration also clearly affect the stability [27]. The instinct stability and structure information of different G-quadruplex sequences may be an obstruction for G4 ligand specific targeting but may provide important hints for instructing the optimization of G4 ligand design.

## 4. Materials and Methods

### 4.1. Materials

DNA oligonucleotides c-*myc* (5′-TGA GGG TGG GGA GGG TGG GGA A-3′, 22 mer) and Telomere G1 (5′-TAG GGT TAG GGT TAG GGT TAG GG-3′, 23 mer) were purchased from Hokkaido System Science Co., Ltd. (Sapporo, Japan). Polyethylene glycol with an average molecular weight of 200 (PEG200) was purchased from Wako Pure Chemical Industries Ltd. (Osaka, Japan) and was used without further purification. Before use, the DNA was annealed under the following conditions: heating to 95 °C for 10 min, then cooled to 25 °C at 0.5 °C/min. **1** was synthesized according to the previously reported research [13].

### 4.2. Synthesis of cNDI Derivatives

*Preparation of **2**:* A suspension of *N*,*N*-bis[3-(3-Aminopropyl)methylaminopropyl]-naphthalene-1,4,5,8-tetracarboxylic acid diimide (0.49 g, 0.50 mmol), Isophthalic acid (0.087 g, 0.53 mmol), 1-[Bis(dimethylamino)methyliumyl]-1H-1,2,3-triazolo[4,5-b]pyridine-3-oxide hexafluorophosphate (0.58 g, 1.5 mmol) was dissolved in a mixture of N,N-dimethylformamide (200 mL) and triethylamine (10 mL), then the solution was stirred at room temperature. After 15 h, the solvent was removed under reduced pressure and dried under vacuum drying. Then, 50 mL of chloroform were added with a small amount of diethylamide to dissolve the solid, and 100 mL of Milli-Q water for extraction. The residual organic phase was applied to silica gel column chromatography (Merck 60) eluted with chloroform/methanol/diethylamine (*v*/*v*/*v* = 1/0.025/0.05). The fraction with Rf of 0.525 in TLC was collected. Pure 2 was obtained as confirmed by a reversed-phase high performance liquid chromatography (HPLC) (Appendix A). Yield: 30.9 mg (9.5%). MALDI-TOF-MASS (positive mode, DHBA) *m*/*z* = 654.5487 (calculated value of C_36_H_40_N_6_O_6_ + H^+^ = 653.7475) (Appendix A); and ^1^H-NMR (500 MHz, CDCl_3_): δ = 1.92–1.97 (8H, *J* = 5.0 Hz, quin.), 2.08 (6H, s), 2.26–2.30 (4H, m), 2.49–2.51 (4H, *J* = 3.3 Hz, t), 3.11–3.14 (4H, *J* = 3.8 Hz, q), 4.31–4.34 (4H, *J* = 5.0 Hz, t), 7.94–7.95 (1H, *J* = 3.3 Hz, t), 8.54 (4H, s), 8.62–8.63 (2H, *J* = 2.5 Hz, d), and 8.69 (1H, s) ppm (Appendix A). ^13^C-NMR (125 MHz, CDCl_3_): δ = 134.9, 130.6, 129.4, 128.7, 126.6, 125.6, 77.3, 77.0, 76.8, 56.7, 55.3, 42.1, 39.5, 38.7, 26.6, 24.9 ppm (Appendix A).

*Preparation of **3**:* A suspension of *N*,*N*-bis[3-(3-Aminopropyl)methylaminopropyl]-naphthalene-1,4,5,8-tetracarboxylic acid diimide (0.98 g, 1.0 mmol), Glutaric acid (0.13 g, 1.0 mmol), 1-[Bis(dimethylamino)methyliumyl]-1H-1,2,3-triazolo[4,5-b]pyridine-3-oxide hexafluorophosphate (1.14 g, 3.0 mmol) was dissolved in a mixture of N,N-dimethylformamide (250 mL) and triethylamine (5 mL), then the solution was stirred at room temperature. After 2.5 h, the solvent was removed under reduced pressure and dried under vacuum drying. Residuals were separated by silica gel column chromatography (Merck 60) eluted with chloroform/diethylamine (*v*/*v* = 1/0.03). The fraction with Rf of 0.25 in TLC was collected. Pure 3 was obtained, as confirmed by a reversed-phase high-performance liquid chromatography (HPLC) (Appendix A). Yield: 170 mg (28%). MALDI-TOF-MASS (positive mode, DHBA) *m*/*z* = 620.3467 (calculated value of C_33_H_42_N_6_O_6_ + H^+^ = 619.7314) (Appendix A); and ^1^H-NMR (500 MHz, CDCl_3_): δ = 1.16 (4H, *J* = 6.75 Hz, quin.), 1.63 (2H, m), 1.86 (14H, m), 2.12 (4H, *J* = 6.50 Hz, t), 2.39 (4H, *J* = 6.11 Hz, t), 2.74 (4H, *J* = 6.47 Hz, q), 4.26 (4H, *J* = 6.74 Hz, t), 6.39 (2H, *J* = 5.02 Hz, t), and 8.67 (4H, s) ppm (Appendix A). ^13^C-NMR (125 MHz, CDCl_3_): δ =170.8, 162.3, 129.7, 125.7, 55.9, 54.2, 40.5, 38.6, 37.0, 34.9, 25.6, 23.7, 20.4 ppm (Appendix A).

### 4.3. Circular Dichroism (CD) Measurement

CD spectra of the annealed 1.5 μM annealed Telomere G1 or c-*myc* was measured in dilute condition (50 mM Tris-HCl buffer (pH 7.4) and 100 mM KCl), and molecular crowding condition (50 mM Tris-HCl buffer (pH 7.4), 100 mM KCl and 40% (*v*/*v*) PEG200), at 25 °C with JASCO J-820 spectrophotometer equipped with a temperature controller, in the presence of 0 μM to 4.5 μM of 1*–*3. CD spectra of un-annealed 1.5 μM Telomere G1 was measured in the cation-deficient crowding condition (50 mM Tris-HCl buffer (pH 7.4) and 40% (*v*/*v*) PEG200), at 25 °C with a JASCO J-820 spectrophotometer equipped with a temperature controller, in the presence of 0 μM to 15 μM of 1–3. The measurement was performed at a scan rate of 50 nm/min, using a Jasco J-820 spectropolarimeter (Tokyo, Japan) with the following conditions: response, 4 s; data interval, 0.2 nm; sensitivity, 100 mdeg; bandwidth, 2 nm; and scan number, 4 times.

### 4.4. Melting Temperature (Tm) Detection

Melting curves of annealed 1.5 μM Telomere G1 and c-*myc* at 288 nm and 263 nm were measured in dilute condition (50 mM Tris-HCl buffer (pH 7.4) and 100 mM KCl for Telomere G1; 50 mM Tris-HCl buffer (pH 7.4) and 5 mM KCl for c-*myc*) and molecular crowding condition (50 mM Tris-HCl buffer (pH 7.4), 1 mM KCl and 40% (*v*/*v*) PEG200 for Telomere G1; 50 mM Tris-HCl buffer (pH 7.4), 0.01 mM KCl and 40% (*v*/*v*) PEG200 for c-*myc*) in the absence or presence of 4.5 μM of **1**–**3**. Melting curves of un-annealed 1.5 μM Telomere G1 was measured in cation-deficient crowding condition (50 mM Tris-HCl buffer (pH 7.4) and 40% (*v*/*v*) PEG200) in the absence or presence of 15 μM of **1***–***3**. The measurement was conducted using a Jasco J-820 spectrophotometer equipped with a temperature controller with the following condition: response; 100 mdeg, temperature gradient; 60 °C/h, response; 1 s; data collecting interval; 0.5 °C, and bandwidth; 1 nm. A total volume of 3 mL was used in the cell with 1 cm of light path length.

### 4.5. Isothermal Titration Calorimetry (ITC) Measurements

ITC measurements were performed using a low volume nano ITC (TA instruments, New Castle, DE, USA) with a cell volume of 190 μL at 25 °C. Annealed DNA solution and chemicals were degassed for 10 min before loading. The measurement was performed with titrating **1**, **2**, or **3** (0–100 μM) to telomere G1 (10 μM) or c-*myc* (10 μM) in 50 mM KH_2_PO_4_-K_2_HPO_4_ buffer (pH 7.0) at 25 °C. For molecular crowding condition, all of the chemical and DNA solution was prepared with a buffer containing 50 mM KH_2_PO_4_-K_2_HPO_4_ buffer (pH 7.0) and 40% (*v*/*v*) PEG200. In each titration, 2 µL of ligand solution was injected into a quadruplex solution every 120 s up to a total of 25 injections, using a computer-controlled 50 µL microsyringe, with stirring at 300 rpm. The binding curve was fitted onto the independent binding model.

### 4.6. UV-Vis Absorption Spectroscopy

The binding affinity of **1**–**3** to annealed Telomere G1 or c-*myc* was studied with Hitachi U-3310 spectrophotometer (Tokyo, Japan). Under a dilute condition: 120 μL of 150 μM telomere G1 or c-myc were added to 5 μM **1**, **2**, or **3** in 50 mM Tris-HCl buffer (pH 7.4) and 100 mM KCl; under a molecular crowding condition: 120 μL of 150 μM telomere G1 or c-*myc* were added to 5 μM **1**, **2**, or **3** in 50 mM Tris-HCl buffer (pH 7.4), 100 mM KCl and 40% (*v*/*v*) PEG200. Absorbance spectra were taken at 25 °C. The observed spectrum changes at 385 nm were rearranged with Scatchard plot [28] as the following equation in the case of non-cooperative binding:*ν*/*L* = *K*(*n*−*ν*)
where *ν*, *L*, *n*, and *K* refer to the saturation fraction as the amount of bound ligand per added DNA (Telomere G1 or c-*myc*), amount of unbound ligand, binding number of ligand per one DNA molecule, and binding affinity, respectively.

## 5. Conclusions

In this study, the interaction of cNDI derivatives with G-quadruplex under molecular crowding conditions were investigated in detail. cNDI derivatives still demonstrated a high ability to recognize and stabilize G-quadruplex under a crowding environment, and the binding affinity was slightly decreased but still comparable to an in-vitro dilute condition. Without the assistance of potassium ion, cNDIs favor telomere G1 to form a hybrid G-quadruplex under molecular crowding condition.

## Figures and Tables

**Figure 1 molecules-25-00668-f001:**
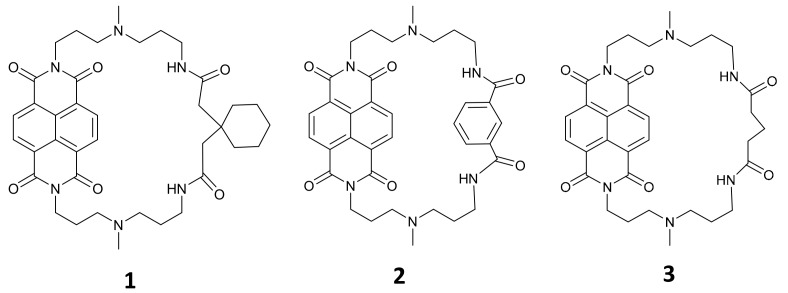
Chemical structures of cyclic naphthalene diimide (cNDI) derivatives **1**–**3**.

**Figure 2 molecules-25-00668-f002:**
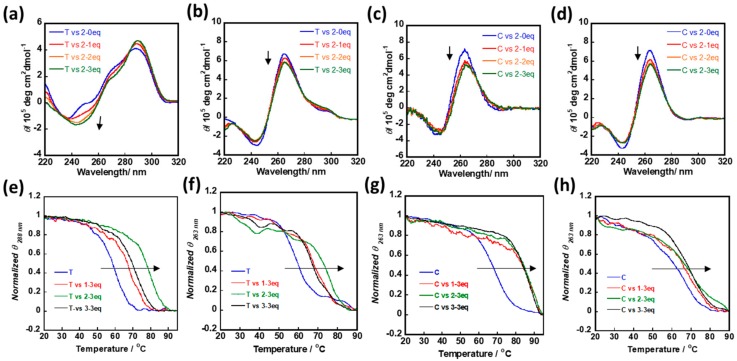
cNDI derivatives recognize and stabilize G-quadruplex under diluting condition and molecular crowding condition. (**a**) **2** recognizes telomere G1 with 50 mM Tris-HCl (pH 7.4) buffer and 100 mM KCl, (**b**) **2** recognizes telomere G1 with 50 mM Tris-HCl buffer (pH 7.4), 100 mM KCl, and 40% (v/v) polymers polyethylene glycol (PEG)200; (**c**) **2** recognizes c-*myc* with 50 mM Tris-HCl (pH 7.4) and 100 mM KCl, (**d**) **2** recognizes c-*myc* with 50 mM Tris-HCl buffer (pH 7.4), 100 mM KCl, and 40% (v/v) PEG 200; (**e**) melting curve of telomere G1 with adding **1**, **2**, or **3** for 3 equivalents with 50 mM Tris-HCl (pH 7.4) and 100 mM KCl; (**f**) melting curve of telomere G1 with adding **1**, **2**, or **3** for 3 equivalents with 50 mM Tris-HCl buffer (pH 7.4), 1 mM KCl, and 40% (*v*/*v*) PEG 200; (**g**) melting curve of c-*myc* with adding **1**, **2**, or **3** for 3 equivalents with 50 mM Tris-HCl buffer (pH 7.4) and 5 mM KCl; (**h**) melting curve of c-*myc* with adding **1**, **2**, or **3** for 3 equivalents with 50 mM Tris-HCl buffer (pH 7.4), 0.01 mM KCl, and 40% (*v*/*v*) PEG 200. T: telomere G1; C: c-*myc*; eq: equivalent.

**Figure 3 molecules-25-00668-f003:**
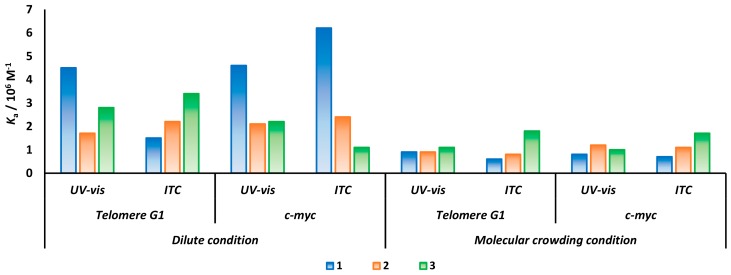
The binding affinity of cNDI derivatives to G-quadruplex under diluting condition and molecular crowding condition based on ITC and UV-vis spectra absorbance.

**Figure 4 molecules-25-00668-f004:**
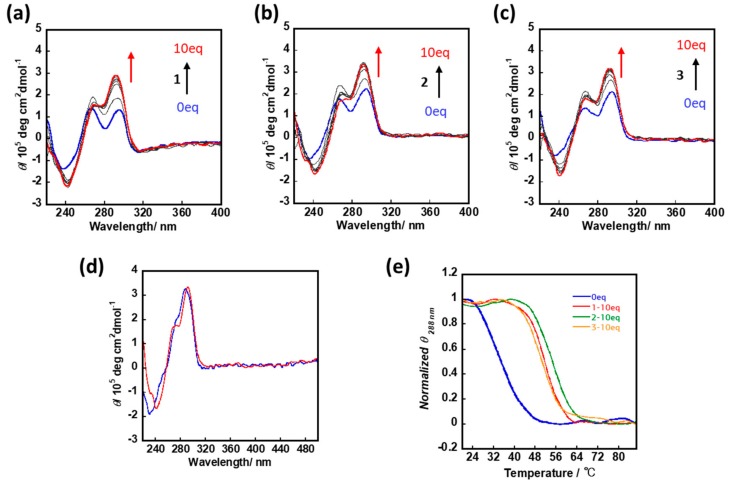
cNDI derivatives induce the formation of hybrid telomere G-quadruplex under cation-deficient molecular crowding condition. Adding (**a**) **1**, (**b**) **2**, or (**c**) **3** to telomere G1 from 0 to 10 equivalents under molecular crowding condition without K^+^ (50 mM Tris-HCl buffer (pH 7.4) and 40% (v/v) PEG 200); (**d**) circular dichroism (CD) spectra of **2**—telomere G1 complex under molecular crowding condition without K^+^ (red), or CD spectra of telomere G1 under diluting condition with 100 mM K^+^ (blue); (**e**) cNDI derivatives enhance the stability of telomere sequence under molecular crowding condition without K^+^.

**Table 1 molecules-25-00668-t001:** cNDI derivatives bind to G-quadruplex under molecular crowding condition.

	Dilute Condition						Molecular Crowding Condition					
	Telomere G1			c-*myc*			Telomere G1			c-*myc*		
	1	2	3	1	2	3	1	2	3	1	2	3
10^−6;^ *K*_a_/M^−1^	1.5	2.2	3.4	6.2	2.4	1.1	0.6	0.7	1.9	0.7	0.8	1.6
*n*	1	2	2	1	2	1	1	1	1	1	1	1
∆*H*/kcal mol^−1^	−12.8	−9	−8.5	−13.9	−10	−11	−43	−35	−22	−20	−53	−20
−*T*∆*S*/kcal mol^−1^	4.4	0.36	−0.4	4.7	1.3	2.9	35	27	14	12	45	12
∆*G*/kcal mol^−1^	−8.4	−8.7	−8.9	−9.2	−8.7	−8.1	−7.9	−8.0	−8.6	−8.0	−8.0	−8.5

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
