# Peer review of "The Interaction of Cyclic Naphthalene Diimide with G-Quadruplex under Molecular Crowding Condition"

_molecules, 2020, doi:10.3390/molecules25030668_

Round 1
Reviewer 1 Report
Tingting Zou and colleagues present in paper named “The interaction of cyclic naphthalene diimide with G-quadruplex under molecular crowding condition” original results briefly describing the interaction of new naphthalene diimide (NDI) derivatives, together with one reported in previous paper of the group (ref. 13), with two selected guanine quadruplexes under molecular crowding conditions. New derivatives reported provide some level of novelty and originality of the results. Targeting the guanine quadruplexes, one of the hot topics of recent biology, ensures the significance of the research.
Unusually, my first major concern relates to English language. The manuscript is poorly written with many mistakes or typos and some sentences are even hard to understand. I strongly recommend detailed text editing and language correction by a native speaker or specialized company.
The scientific concept of the research is correct with no major flaws; however, there is a number of minor points significantly decreasing the quality of the manuscript:
Why the CD spectra of G1 with 0 equivalents of NDI derivatives in fig 4 a-c are different, if they represent the same oligonucleotide in the same conditions? This difference is particularly important because it interfere with the reported effect of ligand interaction on CD of G1, explained as “cNDIs were supposed that could induce telomere to form a hybrid G-quadruplex even under molecular crowding condition.” (l. 136). Is there a reason for showing spectra up to 500 nm (induced CD)? I would also recommend using the same Y-axis for all similar panels in figure.
The authors put special emphasis on the “un-annealed” conditions in experiments with crowding conditions without K+. Does the annealing significantly change the behavior of quadruplexes and their interaction with 1-3?
Why the phosphate buffer system was used for ITC experiments, instead of Tris.Cl used for other experiments? The ITC method settings are not sufficiently described (number / volume of injections, concentration of 1-3 in injector, time between injections, etc).
Is there any particular reason for variable “stoichiometry” (n) of the interactions of G1 with 1-3 ligands or it is just the result of imprecisely determined concentrations during ITC data analysis?
108 – “and Ka of 1 was several folds stronger than 2 and 3.” – this follows from which results? 114 – “consistent binding affinity was acquired from both techniques…” What about G1 with 1 in diluted conditions? The difference between techniques here is bigger than most of the differences between ligands or oligonucleotides.The labeling of oligonucleotides in figure 2 is different from that in text (G1 vs T, c-myc vs C) and other figures.
Instead of / besides the absorption spectra (figs. S10-S11), the titration curves or Scatchard plots should be shown as well.
How was the melting temperature calculated (to get G1 alone Tm 65°C)?
Interestingly, the stabilizing interaction of ligands with parallel quadruplexes is reflected by decreased CD at 264 nm. Very often, the stabilizing effects of proteins or ligands manifest in an opposite way i.e. increased CD at 264 nm. Do you have any explanation for this phenomenon?
In conclusion, although the authors report novel results and the paper concept is correct, the present form of the manuscript is not sufficient for publishing in Molecules without major revision.
Reviewer 2 Report
The manuscript fulfills standard criteria of scientific article. I recommend publishing manuscript as it is. Authors clearly demonstrate the binding effect of three different cNDI ligands with two representative G-rich DNA sequences adopting a stable G-quadruplex motif in dehydrating condition. Experiments are well performed and presented, results are more-less interpreted correctly.
Not necessary recommendation:
i) PEG 200 is dehydrating agent and not an ideal crowding agent due to its size. I just recommend mentioning about it at least in discussion.
ii) Size if inserted legends directly in supplementary Figure S7 is too small. A small augmentation of letter would be acceptable.
Round 2
Reviewer 1 Report
The authors present revised version of the manuscript named “The interaction of cyclic naphthalene diimide with G-quadruplex under molecular crowding condition”. They exhaustively responded to all comments and they somehow incorporated most of them into the manuscript. My main point, the quality of the manuscript text, was significantly improved, in terms of both English language and general readability. The authors added several recommended comments or explanations and they provided all the mentioned missing data like titration curves and Scatchard plots in fig S11, very illustrative Table S1 etc.). In my opinion, the manuscript quality was sufficiently improved and it now meets the standards necessary for publishing in Molecules. I recommend the manuscript for publishing.